# Deep Multi-Layer Perception Based Terrain Classification for Planetary Exploration Rovers

**DOI:** 10.3390/s19143102

**Published:** 2019-07-13

**Authors:** Chengchao Bai, Jifeng Guo, Linli Guo, Junlin Song

**Affiliations:** 1School of Astronautics, Harbin Institute of Technology, Harbin 150000, China; 2China Aerospace Science and Technology Corporation, Beijing 100000, China

**Keywords:** planetary rover, terrain classification, vibration, multi-layer perception, deep neural network, field test

## Abstract

Accurate classification and identification of the detected terrain is the basis for the long-distance patrol mission of the planetary rover. But terrain measurement based on vision and radar is subject to conditions such as light changes and dust storms. In this paper, under the premise of not increasing the sensor load of the existing rover, a terrain classification and recognition method based on vibration is proposed. Firstly, the time-frequency domain transformation of vibration information is realized by fast Fourier transform (FFT), and the characteristic representation of vibration information is given. Secondly, a deep neural network based on multi-layer perception is designed to realize classification of different terrains. Finally, combined with the Jackal unmanned vehicle platform, the XQ unmanned vehicle platform, and the vibration sensor, the terrain classification comparison test based on five different terrains was completed. The results show that the proposed algorithm has higher classification accuracy, and different platforms and running speeds have certain influence on the terrain classification at the same time, which provides support for subsequent practical applications.

## 1. Introduction

With the deepening of deep space exploration missions, planetary rovers have become the main mode of detecting activities [1], requiring rovers to have accurate perception of the environment. On the one hand, it can directly obtain information on the state of the unknown environment. On the other hand, it can provide rich environmental information for the planning system and provide support for the path planning and obstacle avoidance of the rover [2,3,4]. Among them, how to effectively classify and identify the terrain is particularly important, as it directly affects the path selection of the rover, and thus determines whether the detection task can be carried out smoothly. In recent years, this direction has gradually become a research hotspot.

At present, terrain classification based on different sensing modes has been deeply studied, such as vision [5,6,7,8,9,10], radar [11,12], sound [13,14,15], inertial measurement [16,17], and multi-sensor fusion [18,19,20,21,22]. However, for the planetary rover process, the operating environment is highly uncertain, and the sensor is prone to failure. For example, in a strong lighting environment, the visual inspection will be in an unreliable state. In the Mars detection environment where there will be changing dusty weather, it is unfavorable for radar. At the same time, considering the cost of the planetary rover, it is impossible to carry a full set of sensor loads and backups due to the limitation of mass and volume in sensor configuration. Therefore, it is necessary to have a robust sensing mode for the environment to realize terrain classification under different environments, and the vibration sensor is a very good choice. On one hand, the vibration information can reflect the terrain variation characteristics well through the feature representation. On the other hand, the vibration sensor is not easily affected by environmental changes and has certain robustness.

At present, research on terrain classification based on vibration information is still in development, and the related results are mainly completed by the Iagnemma team of MIT and the Weiss team of the University of Tübingen, Germany. Among them, Iagnemma et al. [23] proposed an online terrain parameter estimation method in 2002. By solving the simplified equations of cohesion and internal friction angle, the linear least squares estimator was used to estimate the magnitude and variation of cohesion and internal friction angle in real time. On this basis, in 2004, the visual and vibration based planetary terrain classification and recognition method was first proposed [24]. In this method, the sinking of the planet rover’s wheel is measured based on the vision, the terrain parameters are estimated online based on the tactile sense, and the terrain is classified and identified based on the vibration feedback. Brooks et al. [25] proposed a more complete vibration-based terrain classification method based on Iagnemma in 2005. The offline learning training of the classifier used the labeled vibration data, and the linear classification analysis was used to perform online classification and recognition of the terrain. In 2007, Brooks and Iagnemma [26] proposed a new self-supervised terrain classification method that effectively combined visual and vibration sensing. Firstly, the vibration classification is realized by the acquisition of vibration information in the wheel interaction process. Secondly, the terrain label based on the vibration recognition and the terrain image captured by the vision are learned, so that the visual camera can be used to estimate the front terrain type in the actual application process.

Weiss et al. [27] used support vector machine (SVM) for vibration-based terrain classification in 2006 and proposed a feature extraction method based on radial basis function. Using the unprocessed acceleration data, eight features are obtained through calculation, each feature is normalized and the obtained result is used as a feature vector. A combined feature representation method was also proposed, in which the feature vector composed of the above eight features is simply combined with the feature vector proposed by Brooks into a single feature vector. Later, Weiss et al. [28] proposed techniques for measuring vibration acceleration in different directions to improve classification performance. In this method, the acceleration in the front–rear direction (x-direction), the left–right direction (y-direction), and the up–down direction (z-direction) are measured, each feature is normalized by fast Fourier transform (FFT), and the support vector machine is used for classification. The results show that the classification accuracy of eigenvectors with y-direction acceleration is higher than that of eigenvectors with z-direction acceleration. The classification accuracy with all three is the highest. Later, Weiss et al. [29] realized terrain classification based on the support vector machine and analyzed the influence of different vibration measurement directions on the results, thus giving a simple and effective acquisition mode. In 2008, Weiss et al. [30] proposed a visual/vibration-combined terrain classification method similar to that proposed by Brooks in 2007. The author used visual information to classify the terrain in front of the platform. When the mobile platform reaches the shooting position, the vibration classification result is used to verify the previous classification prediction, so that the prediction compensation can be effectively performed. In 2009, Weiss et al. [31] proposed a classification method based on Bayesian filtering to consider the effects of multiple measurements. Compared with the single measurement classification based on the support vector machine, it can be seen that the recognition effect is obviously improved. Weiss and Zell [32] gave a terrain classification and recognition system based on vibration sensing for the types of terrain that may be untrained when the rover moves in unknown terrain. The system uses the Gaussian mixture model for detection. When the mobile platform collects enough unknown terrain data, this new category will be added to the classification model online, thereby expanding the scope of the application, enabling the mobile platform to have self-learning and self-running capabilities in an unknown environment.

In addition to the above two teams, other scholars have also conducted related research [33,34,35,36]. In 2008, Collins et al. [37] proposed a response-based terrain input classification method. Compared to the existing vibration method, the reliance on speed is reduced by focusing on the traversal of the terrain. This method uses the AGV vibration transfer function to map the vibration output to the terrain input and validate it in the simulation using surface contours from real terrain. In 2012, Tick et al. [38] proposed a multi-level classifier terrain classification method based on angular velocity. The innovation lies in the use of acceleration and angular velocity measurements to characterize features in all basic directions, and the use of sequential forward floating feature selection for feature selection. In 2018, Dupont et al. [39] used probabilistic neural networks to realize online terrain classification and identification of vibration sensing measurements. In the same year, Dupont et al. [40] realized the terrain classification and recognition of mobile platforms at different speeds based on the method of feature space manifold. The vibration sensing measurement unit of the mobile platform was used for data acquisition, and the classification process is combined with principal component analysis to extract and reduce the features, and then the principal component analysis transform coefficients are used to develop and construct the manifold curve. These known coefficients are used to insert unknown coefficients of the terrain as the platform motion speed changes. In 2019, Mei et al. [41] validated the classification effect of different methods based on vibration characteristics. At the same time, it is pointed out that the classification effect of adding a shock absorber is better. Kolvenbach et al. [42] realized classification of planetary soil based on haptic detection. Lomio et al. [43] proposed a method of terrain classification in the indoor environment based on deep learning. Cunningham et al. [44] proposed a thermal inertial measurement method to improve the prediction ability of Mars environmental sideslip. Wang [45] gave a comparative analysis of different sensing modes for terrain classification.

Based on the existing research, combined with the characteristics of planetary rover, this paper proposes a new three-dimensional terrain classification and recognition method based on multi-layer perception. Firstly, based on the Clearpath Jackal unmanned vehicle platform, a vibration sensor is used for data acquisition; secondly, the feature vector for training is obtained by using fast Fourier transform and normalization processing. Then, using the multi-layer perception deep network training proposed in this paper, the mapping relationship between vibration information and terrain types is obtained. Finally, the comparison tests were carried out in different environments, different platforms and different speed environments. The remainder of this paper is organized as follows.

In Section 2, the algorithm framework and feature extraction method are given. Section 3 focuses on the terrain classification and recognition method based on vibration sensing information, which gives detailed explanation from multi-layer perception depth network design. Section 4 discusses the tests and verification based on a mobile platform, and makes comparative analysis of three aspects: the classification accuracy test of the proposed algorithm, the impact of different moving speeds on the classification accuracy, and the impact of different platforms on the classification accuracy. In the end, the conclusion is presented in Section 5.

## 2. Overview

### 2.1. Algorithm Framework

Terrain classification based on vibration information involves three key processes. The first is the collection and processing of data. The second is offline training of the classification model. The last is the online classification identification application. For the acquisition and processing of vibration signals, this paper is based on the Clearpath Jackal unmanned vehicle platform equipped with a three-axis AFT601D vibration sensing unit. As shown in Figure 1, the experimental platform collected and pre-processed signals in five different material topographies, brick, sand, flat, cement, and soil.

After the data was collected, offline training of the deep neural network model was performed. As shown in Figure 2, in the offline training phase, in order to enable the online classification stage to have higher resolution capability for different terrains, the three-dimensional raw vibration data collected by the sensor was first initialized and segmented to form a vector with a duration of one second. Secondly, the frequency domain was transformed by fast Fourier transform, and the eigenvectors for training were obtained by normalization. Finally, a multi-layer perception, deep neural network was built for parameter learning, and a network model that can be used online was obtained, which was then used for online terrain classification prediction.

The core of the method lies in the characterization of vibration information and the training of the multi-layer perception, deep neural network. Therefore, a reasonable feature representation should be adopted for the processing of vibration information, and appropriate network structure, activation function and loss representation should be adopted for deep network training. Therefore, based on the given classification implementation framework, the following focuses on feature extraction and deep neural network design.

### 2.2. Feature Extraction

First of all, it is necessary to represent and extract the characteristics of the vibration signal. Choosing the appropriate representation mode is crucial for the subsequent operation processing. During the training process, it is necessary to learn the vibration signal characteristics of the known terrain type. Therefore, the experimental platform needs to traverse different surfaces multiple times to collect vibration information. This study used three, single-axis, vibration sensing units to achieve a working frequency of 100 Hz. In order to facilitate the processing of the data, the collected vibration data was segmented, and each segment corresponds to a stroke of one second of the experimental platform, so that a vector of one size can be generated, that is:(1)$$DATA_{i}|{}_{i=1,2,3}\to V_{i}^{1\times 100}|{}_{i=1,2,3}=[v_{i1}\;v_{i2}\;\ldots v_{i100}]$$
where $DATA_{i}$ is the vibration data collected on each axis, i=1,2,3 is the x, y, z axis, respectively. $V_{i}$ is a vector consisting of vibration data collected in one second.

Finally, mark each vector as its corresponding terrain type.
(2)$$V_{i}^{1\times 100}|{}_{i=x,y,z}=[v_{i1}\;v_{i2}\;\ldots v_{i100}]\to T_{j}|{}_{j=1,2,3,4,5}$$
where $T_{j}$ is the type of terrain, j=1,2,3,4,5 correspond to brick, sand, flat, cement, and soil. Next, the original vibration signal was converted to the frequency domain. The original data was normalized first, and each vibration vector was normalized to a vector with a mean of 0 and a standard deviation of 1.
(3)$$V_{i}^{*}=\frac{V_{i}-E(V_{i})}{\sqrt{D(V_{i})}}$$
where
(4)$$\begin{array}{l} {E(V}_{i}^{*})=\frac{{E(V}_{i})-E(V_{i})}{D(V_{i})}=0\;\\ {D(V}_{i}^{*})=\frac{{D(V}_{i})}{D(V_{i})}=1 \end{array}$$

Then, a 100 point fast Fourier transform (FFT) was done to convert the original time domain data of the experimental platform into the frequency domain. From the results, this transformation can better characterize the difference between different topography. The reason why the fast Fourier transform was used is that it greatly improves the processing speed compared to the discrete Fourier transform (DFT). For 100 sample points, the fast Fourier transform only needs to be calculated 200 times, while the conventional DFT needs to be calculated 10,000 times. After applying the FFT to each vector, each vector was normalized to the (0,1) interval. Standardization prevents high-level data from being dominated in later training. Thus, after the original data was normalized by the data, the indicators are in the same order of magnitude, which is suitable for comprehensive comparative evaluation.

As mentioned in the survey section, the current vibration-based terrain classification method is mainly based on the vertical measurement, that is, the z-direction, because the terrain change has a large influence in the up and down direction. On this basis, the paper adds the measurement of horizontal data, that is, the front–back and the left–right directions, which increases the dimension of the data, and is intended to find the best representation mode between the multi-dimensional data and the terrain category.

In summary, a simple and effective three-axis, information classification method is presented. In each terrain test phase, the vibration data of the three axes was acquired first, and the vibration data in three directions was obtained through standardization processing, that is, before and after ${(V}_{x}^{1:100})$, left and right ${(V}_{y}^{1:100})$, and up and down ${(V}_{z}^{1:100})$. Then, the normalized signal was transformed by FFT, and thus we got before and after $\left[F{\left(V_{x}\right)}_{1:100}\right]$, left and right $\left[F{\left(V_{y}\right)}_{1:100}\right]$, and up and down $\left[F{\left(V_{z}\right)}_{1:100}\right]$. Finally, each signal was normalized to the (0,1) interval. By connecting the transformed three-axis signal as a feature vector, a feature vector representation for actual training can be obtained. That is
(5)$$F^{*}=[F{\left(V_{x}\right)}_{1:100}\;F{\left(V_{y}\right)}_{1:100}\;F{\left(V_{z}\right)}_{1:100}]$$

At this time, the dimension of the feature vector was 1 × 300. These feature vectors were then trained in neural networks. Through the above method, based on the experimental platform and sensor, the data was collected for five different material topography. As shown in Figure 3, the left picture shows the material used, the middle is the initial data sampling result, and the right picture shows the signal after preprocessing. It can be seen from the results that the conversion of the vibration signal to the frequency domain analysis can better reflect the difference of different materials, and the characteristics of different materials can be distinguished.

## 3. Multi-Layer Perception Deep Neural Network Design

### 3.1. Deep Neural Network

Feedforward neural networks are a special form of supervised neural networks that use high-precision approximate computational models through advanced parallel hierarchical processing structures. Each layer consists of a number of neurons with different functions, and each neuron between the layers is in a fully connected form. Multi-layer perception is a special kind of feedforward neural network, and its data processing is usually carried out in the input layer, hidden side, and output layer. This is similar to the forward propagation of the BP neural network. The difference is that multi-layer perception can design and classify complex data through the design of parameters such as network layer number, number of neurons, and activation function.

Based on the above analysis, this paper designed a five-layer, multi-layer perception, deep neural network to verify the classification and recognition ability of the algorithm for different terrains. Each layer is fully connected. As shown in Figure 4, the first four layers of activation functions are Rectified Linear Units (ReLU) functions [46], and the last layer is the Softmax function [47]. The network input is the eigenvector obtained by preprocessing. The acquisition method is given in Section 2.2. Different materials correspond to different input signals. The output value is then obtained through the processing of the hidden layer, and the output value is the classification of the corresponding terrain.

The calculation transfer instructions are given below. First, given the first layer is the input layer, the input information is $\left[v_{1},v_{2},\ldots ,v_{n}\right]$. For layer l, the number of neurons in each layer is $L_{l}$, the corresponding output is $y_{l}$, the output of the i node is $y_{l}^{\left(i\right)}$, and the corresponding input is $u_{l}^{\left(i\right)}$. The weight matrix connecting the l and l−1 layers is $W_{l}$. The weight from the j node of the l−1 layer to the i node of the l layer is $w_{l}^{\left(ij\right)}$. Then the transfer relationship of the forward propagation calculation can be obtained. That is
(6)$$y_{l}^{\left(i\right)}=f(u_{l}^{\left(i\right)})$$
(7)$$u_{l}^{\left(i\right)}=\sum_{l\in L_{l-1}}w_{l}^{\left(ij\right)}y_{l-1}^{\left(j\right)}+b_{l}^{\left(i\right)}$$
(8)$$Y_{l}=f(U_{l})=f(W_{l}y_{l-1}+b_{l})$$

In which, f(·) is the activation function, $b_{l}^{\left(i\right)}$ is the offset of the j node of the l layer. Compared with the traditional BP neural network, the network design has made corresponding improvements in the selection of the activation function and the calculation of the loss function.

### 3.2. Activation Function Selection

Whether in the BP neural network or in the feedforward neural network, the sigmoid type function is often used as the activation function. The function curve of the sigmoid function and its gradient curve are shown in Figure 5. It can be clearly seen that after the signal is processed by the sigmoid function, a value between (0,1) is output. The part at both ends is compressed to the vicinity of the edge value, and thus leads to the disappearance of the gradient, which has the opposite effect on the rapid convergence of network training.

In order to avoid this phenomenon, this section introduces the Rectified Linear Units (ReLU) into the multi-layer network training. Due to its better gradient characteristics, it plays a positive role in the convergence of deep networks. The ReLU function is actually a piecewise function defined as:(9)ReLU(x)=max{0,x}={xx≥00x<0

As shown in Figure 6, the gradient of the ReLU function is constant, which is more biologically similar to the neuron information transfer mechanism and has a greater efficiency advantage in deep network calculations due to its simplified form. Furthermore, compared with the sigmoid function, the gradient value of the ReLU function is 0 or 1, which solves the problem of gradient disappearance to some extent and improves the speed of network convergence. Finally, because of the segmented nature of the ReLU function, some neurons can be set to output 0 during data processing, which ensures that the network has a certain sparsity and reduces the risk of over-fitting. It has certain optimization effects on network learning ability.

In addition, in order to better distinguish the type of terrain recognition, the Softmax function was introduced after full connection. This function, also known as the exponential normalization function, is a normalized form of a statistical function that maps a K-dimensional real vector to a new K-dimensional real vector in the range (0,1). Its function form is
(10)$$\sigma {\left(Y\right)}_{j}=\frac{e_{j}^{Y}}{\sum_{i=1}^{K}e_{i}^{Y}}\;\;(j=1,2\ldots ,K)$$

The sum of the mapped values is 1, satisfying the nature of the probability. In turn, this thought transformation can be utilized. When the output node is finally selected, the node with the highest probability can be selected, that is, the output value is the largest value. As shown in Figure 7, this function is introduced in the output layer to match the given terrain type by the corresponding value.

### 3.3. Loss Function Selection

The loss function is an important part of the artificial neural network and is used to characterize the difference between the measured estimate and the true value. The result is a non-negative value, and the robustness of the network model increases gradually as the loss function decreases, and conversely, it decreases as the loss function increases. So, choosing the appropriate loss function definition is very important for the training of the network model. In general, the structural risk function of a model can be characterized by empirical risk terms and regularization terms, as follows:(11)$$\begin{array}{ll} \xi^{*} & =\arg\underset{\xi }{\min}L(\xi )+\lambda \cdot \Phi (\xi )\\ & =\arg\underset{\xi }{\min}\frac{1}{n}\sum_{i=1}^{n}L(y^{\left(i\right)},f(x^{\left(i\right)},\xi ))+\lambda \cdot \Phi (\xi ) \end{array}$$
where Φ(ξ) is a regular term, ξ is the parameter of the model learning, f(·) is the activation function, $x^{\left(i\right)}$ is the training sample data. For the research in this paper, focusing on the empirical risk items, there is
(12)$$L(\xi )=\frac{1}{n}\sum_{i=1}^{n}L(y^{\left(i\right)},f(x^{\left(i\right)},\xi ))$$

In the BP neural network compared with this paper, we use the L2 loss, that is, the mean square error (MSE), to evaluate the deviation between the predicted value and the actual value. This method is often used for regression problems. However, for terrain classification and recognition, the output is a discrete variable, so this section uses the cross-entropy loss function [48] to achieve the same effect. This loss function is often used for binary classification, and the classification for multiple classes can be given by the following function.
(13)$$L=-\frac{1}{n}\sum_{i=1}^{n}\left[y^{\left(i\right)}\log(f(x^{\left(i\right)},\xi ))+(1-y^{\left(i\right)})\log(1-f(x^{\left(i\right)},\xi ))\right]$$

Therefore, the difference between the two probability distributions is measured by cross entropy. If the cross entropy is large, this means that the difference between the two distributions is large. Conversely, if the cross entropy is small, the two distributions are similar to each other.

## 4. Results and Discussion

Based on the above methods, this paper verifies the environment in five terrains. The experimental parameters are set as shown in Table 1. Firstly, the classification accuracy of proposed algorithm and BP Neural Network [49] is compared. Then the influence of the platform on the classification recognition effect under different running speeds is given synchronously. Finally, the effects of different platforms on classification recognition are compared. Furthermore, the advantages and disadvantages of the proposed algorithm in the context of platform and state changes are analyzed from multiple perspectives.

This test collects vibration data for 2 min for different terrain. According to the above segmentation method, 120 sets of data are collected for each type of terrain, so a total of 600 sets of data are collected. In the training process, 500 sets of data are selected randomly, and the remaining 100 sets of data are used as test samples.

Before the analysis, the training curve of the trained multi-layer perception network model on the test data set is given. As shown in Figure 8, the overall prediction accuracy on the test set after the network convergence is 0.883, because the data acquisition area is an outdoor natural environment, which is lower than the laboratory test accuracy, but the accuracy meets the needs of practical applications.

### 4.1. Terrain Test

Using the trained network model, the paper conducted tests in five different material terrain environments. Here, in order to ensure that no other interference was introduced, the mobile platform ran at a constant speed. At the same time, for the convenience of representation, the brick, sand, flat, cement, and soil terrain are represented by the numbers 1, 2, 3, 4, and 5, respectively. Through experiments, the correct classification rates of five types of terrain were obtained, as shown in Table 2.

As can be seen from the above table, the algorithm has higher recognition accuracy for flat terrain, cement terrain, and soil terrain, and lower classification accuracy for brick terrain and sand terrain, which is related to the small discrimination between the two terrain vibration characteristics in the actual data acquisition process. In addition, compared with the BP neural network-based terrain classification results, as shown in Figure 9, it can be clearly seen that the proposed method has a good improvement in the recognition accuracy of each category, especially for the sand environment it has a certain improvement. Through experiments, it can be found that the regularization of a single experimental environment and the discrimination of its various experimental environments have a great influence on the recognition accuracy. This paper starts from the actual environment and chooses the common application terrain. It is intended to improve the adaptability of the algorithm to the environment through the improvement of the algorithm.

In general, Category 1 and Category 4 have lower classification accuracy than other categories. By comparing the actual environment, some parts of the brick were damaged and smoothed, which may be the reason why the feature distinguishing degree is similar to the cement terrain. The sand terrain experiment process was not carried out on lush lawn and the soil terrain environment was carried out on land where there was not only soil, but also weeds. This also causes confusion between the two types of terrain.

In summary, it can be seen from the test results that the recognition accuracy of the terrain based on the deep, multi-layer perception, neural network is better than that based on the BP neural network.

### 4.2. Speed Change Test

It can be seen from the previous section that the algorithm proposed in this paper has a good classification effect on five terrains under constant velocity motion. Based on this, this section analyzes the accuracy of terrain classification under different speed states. In this experiment, the mobile platform, vibration sensing equipment and five terrain environments were the same as in the previous experiment. Data acquisition was performed under v=0.2 m/s, v=0.4 m/s and v=0.6 m/s uniform motion, and five sets of experimental samples were taken at each speed state. The classification accuracy results are shown in Table 3. It can be seen that there has good classification accuracy for five terrains under three speed states.

In order to analyze the influence of the change of speed on the accuracy of terrain classification, the above results are plotted as a histogram, as shown in Figure 10. Overall, speed has no consistent effect on classification accuracy, but it can be found from some details that there is a certain reference value for the improvement of precision. First of all, for the flat and sand environments, with the increase of speed, the classification accuracy increases. This can lead to the conclusion that the higher the speed, the greater the difference between the two terrains and other terrains, the more obvious the feature differentiation degree. Secondly, for brick, cement, and soil environments, the highest classification accuracy is at a speed of v=0.4 m/s, which indicates that for these three terrains, there is no simple relationship where the higher or lower the speed, the higher the classification accuracy and the analysis needs through testing to further examine this. Finally, from the horizontal comparison at different speeds, it can be found that the overall classification accuracy is higher than the other two operating states at a speed of v=0.4 m/s, and the overall recognition accuracy reaches 92.29%. Therefore, from a comprehensive perspective, the speed of the experimental test environment has a certain impact on the classification accuracy of the terrain. It is necessary to perform adaptive analysis according to different platforms and different operating environments.

Similarly, in order to compare the accuracy effect of the deep, multi-layer perception, neural network on terrain classification, the overall classification accuracy based on BP neural network was calculated synchronously. The result is shown in Figure 11. It can be seen that at different speeds, the deep, multi-layer perception, neural network is significantly better than the latter, and from the overall trend, the above analysis of the impact of speed variation on classification accuracy is verified.

### 4.3. Different Platform Test

By analyzing the above test results, it is considered that different platforms have different vibration characteristics due to their own quality and performance, even in the same operating state and the same test environment. Therefore, this section will compare and analyze the impact on terrain classification accuracy for different mobile platforms. The experimental sensor and experimental environment remain unchanged. The mobile platform used the Blue Whale XQ unmanned vehicle platform [50] in addition to the Clearpath Jackal unmanned vehicle platform [51] previously used. At the same time, taking into account the influence of speed, the test was performed at speeds of v=0.2 m/s, v=0.4 m/s and v=0.6 m/s. The classification accuracy on the five terrains is shown in Table 4. It can be seen that good recognition accuracy can be obtained.

Figure 12 shows the histogram of the classification results of the XQ unmanned vehicle platform. It can be seen that the overall classification accuracy of the XQ unmanned vehicle platform has reached a very high level, and there is no uniform law on the influence of speed on its classification accuracy. However, from the horizontal comparison of speed, it has a higher overall classification accuracy at a speed of v=0.4 m/s, which is similar to the results of the Jackal unmanned vehicle platform. It can be inferred that in the terrain environments used in this experiment, the data collected by the test platform at a speed of v=0.4 m/s has an easily distinguishable vibration characteristic, which can increase the recognition accuracy of the terrain.

Figure 13 shows the comparison of the classification and recognition accuracy of the two platforms at different speeds. It can be clearly seen that the recognition accuracy of the XQ platform is significantly better than that of the Jackal platform, and there are opposite results in only a few operating conditions. It can be seen that for the material topography tested in this paper, XQ is more suitable for classification and recognition of terrain. By comparing the parameters between the platforms, the reason for the difference between their results is likely to due to the fact that the XQ platform is lower quality and is more sensitive to terrain changes, and the vibration sensor unit data changes more clearly, which increases differentiation between different terrains, and thus results in better classification results.

Figure 14 shows the comparison between the proposed algorithm and the BP neural network. It can be seen that the test results based on the deep, multi-layer perception, neural network are better than the BP neural network-based classification results under different platforms and different speed conditions. Once again, the advantages of the proposed algorithm over the previous method are illustrated. However, it is also clear to see the impact of different platforms on the classification results. The classification accuracy of the XQ unmanned vehicle platform based on BP neural network is better than that of the Jackal unmanned vehicle platform based on the deep, multi-layer perception, neural network at v=0.2 m/s and v=0.6 m/s. Therefore, the choice of platform is equally important for terrain classification and recognition. On the contrary, for the planetary rover mission, given the configuration of the rover system, it is necessary to select a vibration sensor with the appropriate range to make the feature distinction more obvious and facilitate the development of the task.

## 5. Conclusions

In this paper, a terrain classification and recognition method based on a multi-layer perception, deep neural network is designed for the effective identification of terrain in the process of planetary exploration. Five different outdoor environments have been comprehensively tested based with the Clearpath Jackal unmanned vehicle platform, the Blue Whale XQ unmanned vehicle platform and a vibration sensor. It can be concluded from the results that the terrain estimation algorithm proposed in this paper has better classification and recognition accuracy for the five terrains tested, and the algorithm has certain advantages compared with the traditional BP neural network-based classification results. Furthermore, through comparison experiments at different speeds and different platforms, it can be concluded that the choice of the experimental platform and the difference in operating speed will have an impact on the recognition results. Therefore, in the actual application process, it is necessary to select a suitable platform according to different requirements.

In addition, the proposed method needs offline learning and training, which is very difficult for planetary exploration as it is hard to obtain real terrain vibration data, especially for the first exploration area, as it is impossible to obtain in advance. To overcome this limitation, a feasible model is as follows: (1) Based on the method proposed in this paper, the rover can distinguish different soft and hard terrain materials. (2) Sampling data taken from the safe visual range of the landing area for planetary exploration, the classification ability of existing networks is improved based on sampled small sample data. (3) Improving the self-learning ability and robustness of the network to new data based on transfer learning so it can provide feasible support for practical application.

In summary, the research in this paper has certain significance, but it is still necessary to expand the research and analysis of different material topography, and to expand the analysis of the speed tests within the allowable range of the platform, so as to give more general experimental conclusions.

## Figures and Tables

**Figure 1 sensors-19-03102-f001:**
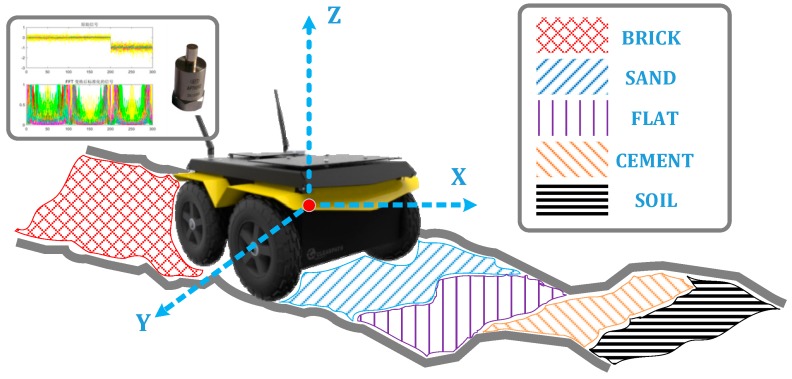
Experimental flow diagram.

**Figure 2 sensors-19-03102-f002:**
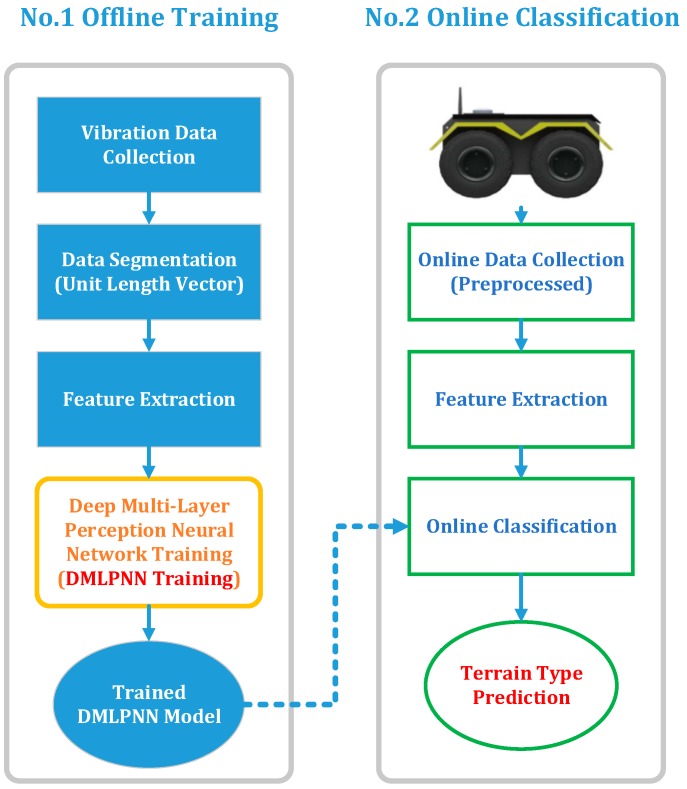
Neural network classification flow chart.

**Figure 3 sensors-19-03102-f003:**
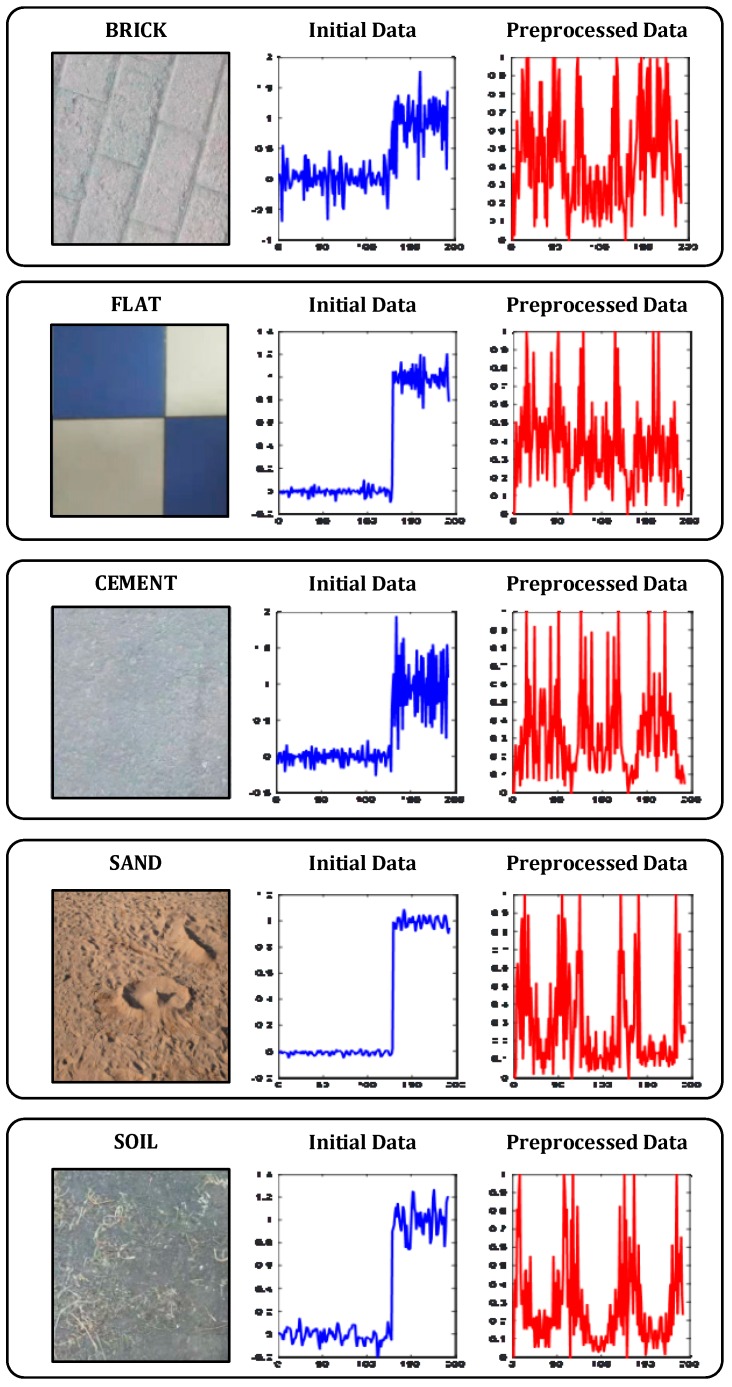
Data acquisition results.

**Figure 4 sensors-19-03102-f004:**
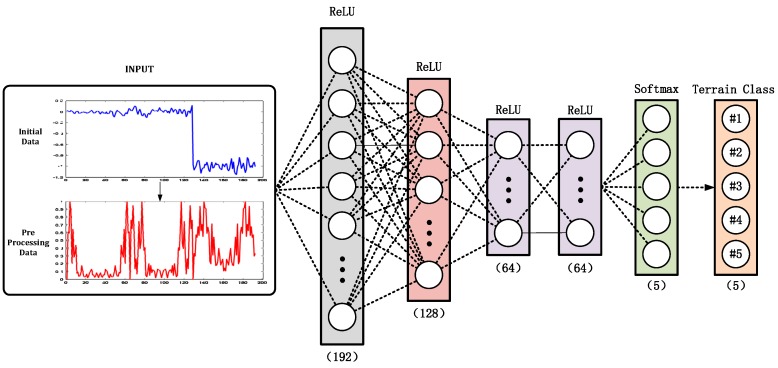
Multi-layer perception, deep neural network structure. ReLU: Rectified Linear Units.

**Figure 5 sensors-19-03102-f005:**
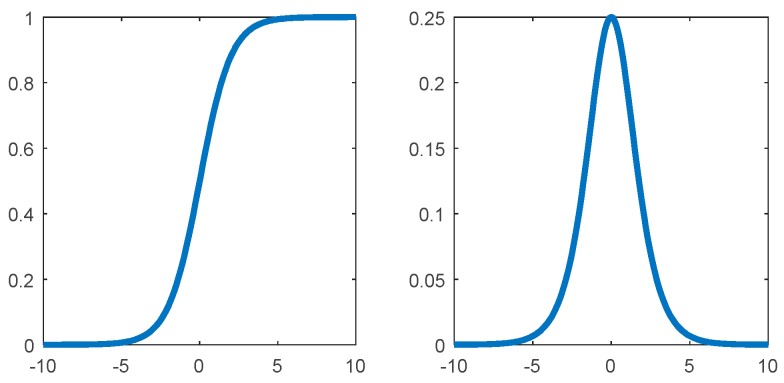
Sigmoid type activation function and its gradient map.

**Figure 6 sensors-19-03102-f006:**
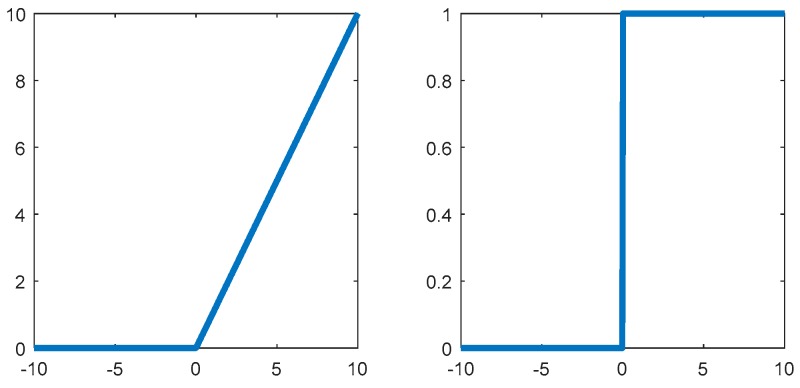
ReLU function and its gradient map.

**Figure 7 sensors-19-03102-f007:**
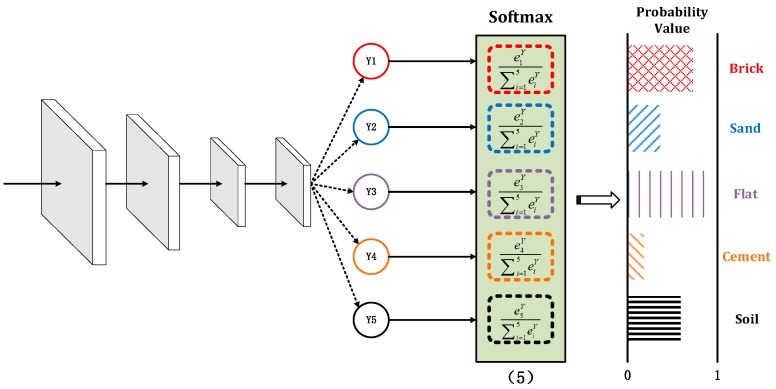
Softmax function output.

**Figure 8 sensors-19-03102-f008:**
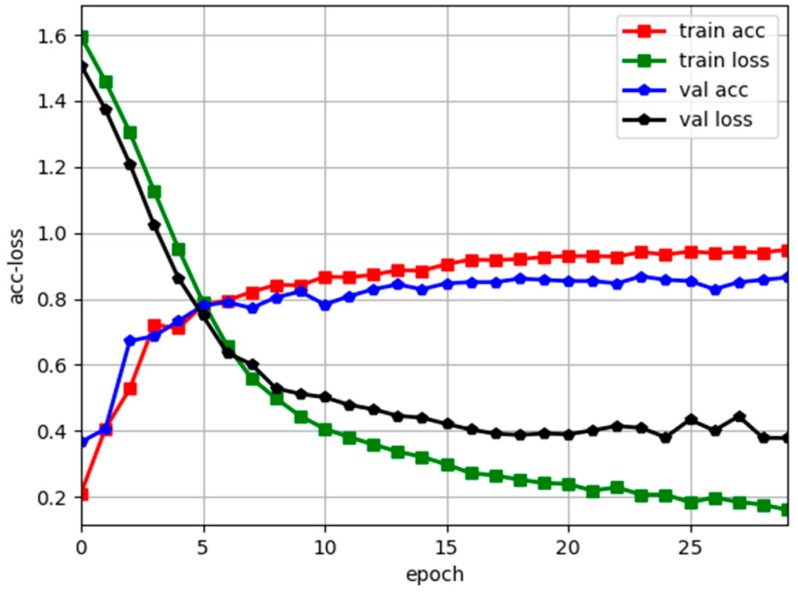
Training results.

**Figure 9 sensors-19-03102-f009:**
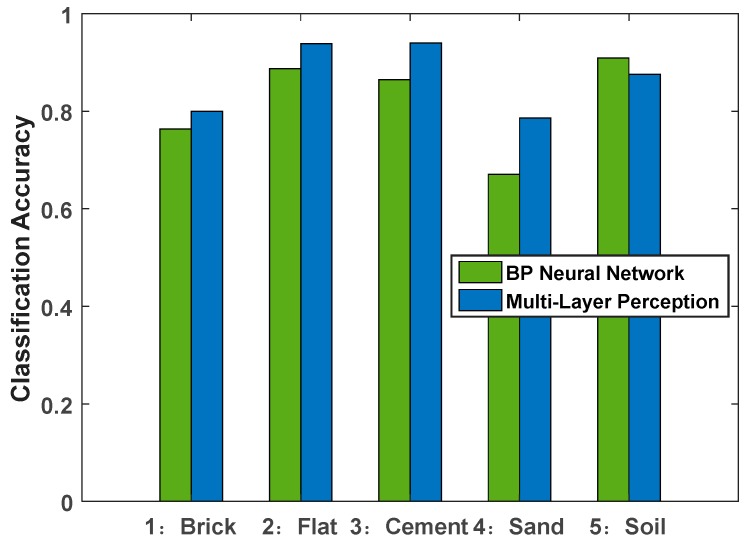
Comparison of prediction accuracy of different algorithms.

**Figure 10 sensors-19-03102-f010:**
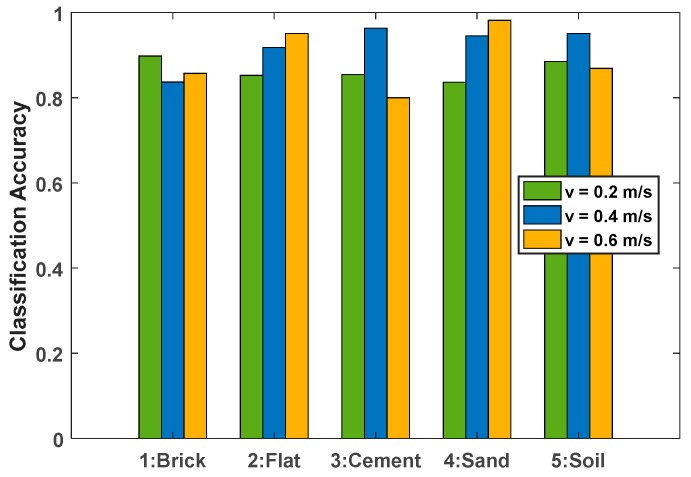
Classification accuracy of five types of terrain at different speeds.

**Figure 11 sensors-19-03102-f011:**
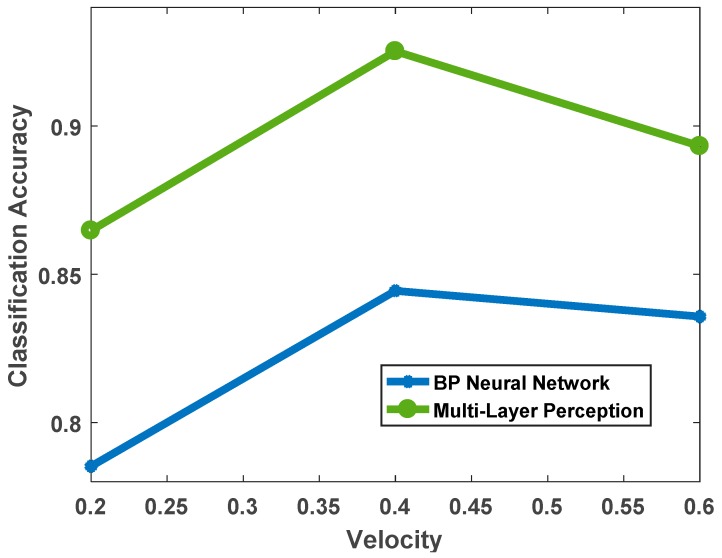
Classification accuracy of two algorithms at different speeds.

**Figure 12 sensors-19-03102-f012:**
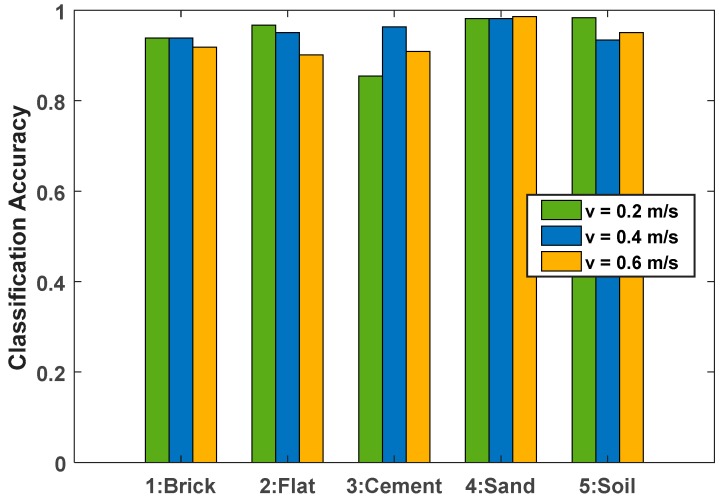
Classification accuracy based on the XQ platform at different speeds.

**Figure 13 sensors-19-03102-f013:**
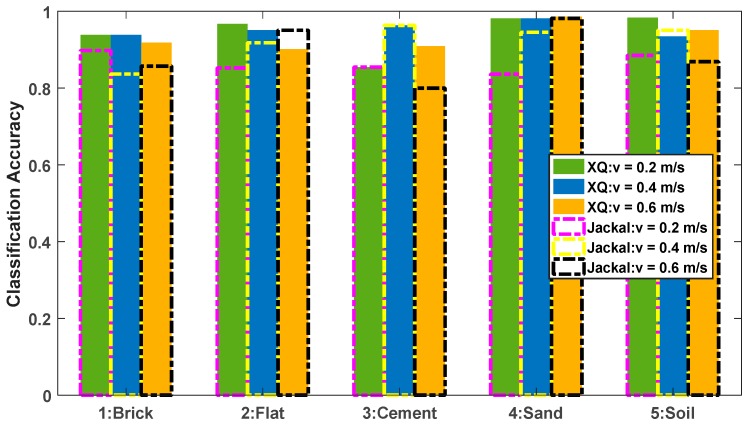
Accuracy comparison of different platforms at different speeds.

**Figure 14 sensors-19-03102-f014:**
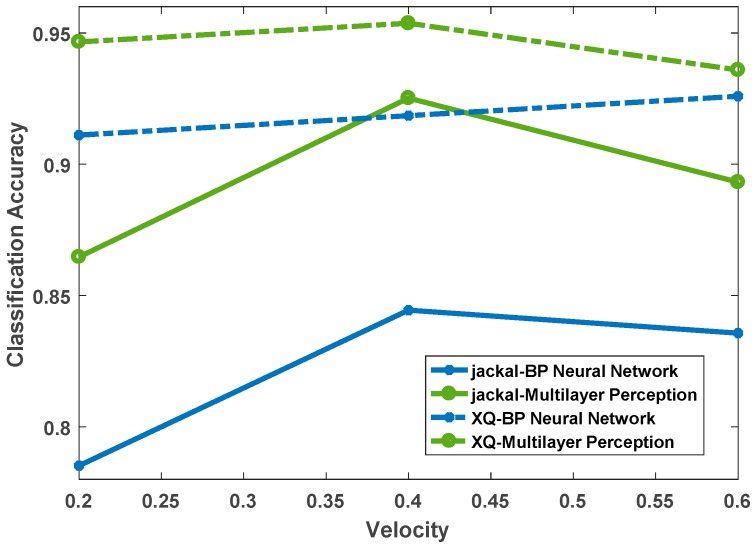
Classification accuracy of two algorithms at different speeds.

**Table 1 sensors-19-03102-t001:** Setting of experimental parameters.

Parameters	#1	#2	#3	#4	#5
Test Terrain	brick	sand	flat	cement	soil
Collect Data	120 sets	120 sets	120 sets	120 sets	120 sets
Total Data	600 sets (Training samples: 500 sets; Test samples: 100sets)				
Test Velocity	0.2 m/s, 0.4 m/s, 0.6 m/s				
Test Platform	Clearpath Jackal, XQ				

**Table 2 sensors-19-03102-t002:** Prediction accuracy of different categories for different experiments.

Experiment	Accuracy #1	Accuracy #2	Accuracy #3	Accuracy #4	Accuracy #5
1	0.8571	0.9487	0.9500	0.7674	0.9459
2	0.7857	0.9487	0.9250	0.7906	0.8918
3	0.7500	0.9230	0.9750	0.8372	0.8108
4	0.7857	0.9230	0.9250	0.7674	0.8648
5	0.8214	0.9487	0.9250	0.7674	0.8648

**Table 3 sensors-19-03102-t003:** Prediction accuracy of different speeds for different terrains.

Experiment	Accuracy #1	Accuracy #2	Accuracy #3	Accuracy #4	Accuracy #5
v=0.2 m/s	0.8979	0.8524	0.8545	0.8363	0.8852
v=0.4 m/s	0.8367	0.9180	0.9636	0.9454	0.9508
v=0.6 m/s	0.8571	0.9508	0.8000	0.9818	0.8688

**Table 4 sensors-19-03102-t004:** Prediction accuracy results.

Platform	Speed	Accuracy #1	Accuracy #2	Accuracy #3	Accuracy #4	Accuracy #5
Jackal	v=0.2 m/s	0.8979	0.8524	0.8545	0.8363	0.8852
	v=0.4 m/s	0.8367	0.9180	0.9636	0.9454	0.9508
	v=0.6 m/s	0.8571	0.9508	0.8000	0.9818	0.8688
XQ	v=0.2 m/s	0.9387	0.9672	0.8545	0.9818	0.9836
	v=0.4 m/s	0.9387	0.9508	0.9636	0.9818	0.9344
	v=0.6 m/s	0.9183	0.9016	0.9090	0.9859	0.9508

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
