# Peer review of "Deep Multi-Layer Perception Based Terrain Classification for Planetary Exploration Rovers"

_sensors, 2019, doi:10.3390/s19143102_

Reviewer 1 Report

Abstract

It presents information on the problem addressed, the proposal formulated, and the conclusions were presented coherently.

1. Introduction

It presents the contextualization of the problem and with models that have already addressed the problem solving with artificial intelligence.

The introduction is very extensive. Reduce text size.

2. Overview

Please explain all the variables of equation 1. In particular the value v.

3. Multi-layer Perception Deep Neural Network Design

Quotes on some themes are missing, for example, the ReLU activation function.

5. Conclusion

Include future work into completion.

General

The models presented are explicit. I have not seen any citations for more recent work.

Please explain the related work in 2019.

The configurations of the executed tests could be presented in graphs to facilitate the reading of the configurations.

Author Response

Dear reviewer,

Thank you for allowing a modification of our manuscript, with an opportunity to address the reviewer’s comments.

We are uploading (a) our point-by-point response to the comments (PDF document), (b) an updated manuscript with yellow highlighting indicating changes (PDF  document). I hope I can get your approval.

Thank you for your patience in reviewing the manuscript.

Best regards,

Chengchao Bai, Jifeng Guo, Linli Guo, Junlin Song

Reviewer 2 Report

The paper under review is proposing deep neural network for terrain classification based on vibration data. The paper is written properly, but it lacks substantial push of the state of the art.

Therefore, the novelty of the paper is rated low. Besides this major problem there are some other issues:

-- "Standardization prevents high-level data from dominated in later training. Thus, after the original data is normalized by the data, the indicators are in the same order of magnitude, which is suitable for comprehensive comparative evaluation." In reviewers opinion with the normalization the problem of high-level data is not removed. The large values still will be large in comparison to small values in the vector.

-- In the paper tests of the state-of-the-art methods presented in the literature review are missing. Authors only compared two approaches proposed by themselves.

-- There is also a question why the authors are having Planetary Exploration Rovers when non of the presented robotic platforms resembles one. Additionally, the test were not performed for the terrain types which could be met in planetary missions.

Author Response

Dear reviewer,

Thank you for allowing a modification of our manuscript, with an opportunity to address the reviewer’s comments.

We are uploading  our point-by-point response to the comments (PDF document). I hope I can get your approval.

Thank you for your patience in reviewing the manuscript.

Best regards,

Chengchao Bai, Jifeng Guo, Linli Guo, Junlin Song

Reviewer 3 Report

This paper attempts to address an ongoing problem with planetary explore vehicles - that of insufficient reliable sensors to ensure effective domain identification in all likely scenarios.  This is a very worthwhile objective and is approached in an appropriate way.  My concerns are how the training data is collected and used in a real mission.  Is it collected in the laboratory or once the vehicle is deployed?  This could be explained in an additional paragraph in the conclusions.

The level of English is very good, but it occasionally slips, particularly in the sections highlighted in yellow.  This needs revising to make the standard consistent.

Several of the bar charts have the key placed over the tops of the bars of the right hand result (normally 'soil').  These need revising by moving the key down so that the height of each bar is clearly visible.

Author Response

Dear reviewer,

Thank you for allowing a modification of our manuscript, with an opportunity to address the reviewer’s comments.

We are uploading  our point-by-point response to the comments (PDF document). I hope I can get your approval.

Thank you for your patience in reviewing the manuscript.

Best regards,

Chengchao Bai, Jifeng Guo, Linli Guo, Junlin Song

Round  2

Reviewer 2 Report

Dear authors,

any of my comments was addressed in the the present revision

Author Response

Thank you for your recognition of this paper.